# Systemic Influences of Mammary Cancer on Monocytes in Mice

**DOI:** 10.3390/cancers14030833

**Published:** 2022-02-07

**Authors:** Amy Robinson, Matthew Burgess, Sheila Webb, Pieter A. Louwe, Zhengyu Ouyang, Dylan Skola, Claudia Z. Han, Nizar N. Batada, Víctor González-Huici, Luca Cassetta, Chris K. Glass, Stephen J. Jenkins, Jeffery W. Pollard

**Affiliations:** 1MRC-Centre for Reproductive Health, Queen’s Medical Research Institute, University of Edinburgh, Edinburgh EH16 4TJ, UK; amyclarerobinson@gmail.com (A.R.); sheila.webb@ed.ac.uk (S.W.); Luca.cassetta@ed.ac.uk (L.C.); 2Centre for Inflammation Research, Queen’s Medical Research Institute, University of Edinburgh, Edinburgh EH16 4TJ, UK; matthew.burgess@ed.ac.uk (M.B.); Pieter.Louwe@ugent.be (P.A.L.); 3Department of Cellular & Molecular Medicine, University of San Diego, LA Jolla, CA 92037, USA; z5ouyang@ucsd.edu (Z.O.); phageghost@gmail.com (D.S.); czh002@health.ucsd.edu (C.Z.H.); cglass@health.ucsd.edu (C.K.G.); 4Institute of Genetic and Molecular Medicine, University of Edinburgh, Edinburgh EH4 2XU, UK; Nizar.batada@gmail.com (N.N.B.); victor.gonzalez@irbbarcelona.org (V.G.-H.)

**Keywords:** breast cancer, monocytes, mouse, human, myeloid

## Abstract

**Simple Summary:**

Using a mouse model of breast cancer driven by the mammary epithelial expression of the polyoma middle T oncoprotein in which the tumors progress from benign to malignant metastatic stages, we show that cancer causes an increase in circulating monocytes and a splenomegaly. This increase in monocyte number is due to their increased proliferation in the bone marrow and not turnover rates in the blood. Single cell sequencing also shows that new populations of monocytes do not arise during cancer. Cancer also drives systemic changes in the monocyte transcriptome, with a notable down-regulation of interferon signaling. These systemic influences start in the bone marrow but intensify in the blood. Comparison of cancer prone and cancer resistant mouse inbred strains carrying the same oncogene reveals that the genetic background of the strain causes different monocyte transcriptional changes. Similarly, a comparison of the mouse transcriptome to human breast cancer monocyte profiles indicates limited similarities, to the extent that interferon signaling is enhanced in humans. Systemic responses are different in the same model of cancer on different genetic backgrounds within a species and even greater changes are found across species. These data suggest that at the very least this mouse model will be limited when it comes to exploring the mechanism behind systemic changes in humans.

**Abstract:**

There is a growing body of evidence that cancer causes systemic changes. These influences are most evident in the bone marrow and the blood, particularly in the myeloid compartment. Here, we show that there is an increase in the number of bone marrow, circulating and splenic monocytes by using mouse models of breast cancer caused by the mammary epithelial expression of the polyoma middle T antigen. Cancer does not affect ratios of classical to non-classical populations of monocytes in the circulation nor does it affect their half-lives. Single cell RNA sequencing also indicates that cancer does not induce any new monocyte populations. Cancer does not change the monocytic progenitor number in the bone marrow, but the proliferation rate of monocytes is higher, thus providing an explanation for the expansion of the circulating numbers. Deep RNA sequencing of these monocytic populations reveals that cancer causes changes in the classical monocyte compartment, with changes evident in bone marrow monocytes and even more so in the blood, suggesting influences in both compartments, with the down-regulation of interferon type 1 signaling and antigen presentation being the most prominent of these. Consistent with this analysis, down-regulated genes are enriched with STAT1/STAT2 binding sites in their promoter, which are transcription factors required for type 1 interferon signaling. However, these transcriptome changes in mice did not replicate those found in patients with breast cancer. Consequently, this mouse model of breast cancer may be insufficient to study the systemic influences of human cancer.

## 1. Introduction

Monocytes are key players in the innate immune system, surveying the vasculature in the steady state or being recruited to normal tissues and to sites of infection or tissue damage where they terminally differentiate into macrophages and dendritic cells [1]. The current consensus is that blood monocytes largely derive from hematopoietic stems cells (HSC) in the bone marrow (BM). These progenitors differentiate through several steps to give the restricted erythro-myeloid progenitors that are present in the more abundant Lin^−^Sca-1^−^c-Kit^hi^ (LK) population that in turn can be further sub-divided into the restricted Granulocyte/monocyte (GMP) progenitor. This bi-potent progenitor then differentiates to a macrophage dendritic cell progenitor (MDP) and, subsequently, the unipotent monocyte progenitor (cMoP) [2]. However, recent evidence suggests there may be alternative pathways directly from the GMP or even directly from HSCs [3,4]. Newly formed pre-monocytes proliferate within the BM [5] before being released into the peripheral circulation in a CCR2-dependent fashion [6]. In mice and humans, there are two dominant blood populations, the classical (Ly6C^hi^ CCR2^hi^ and CD14^hi^ CCR2^hi^ CD16^−^ respectively) and non-classical populations (Ly6C^lo^ CCR2^−^ and CD16^hi^ CCR2^−^ CD14^dim^ respectively) [7,8], with an intermediate population with mixed markers (Ly6C^mid^ and CD14^hi^CD16^hi^ respectively) [9,10]. Studies determining half-lives through pulse labelling DNA synthesis with nucleotide analogues indicate that classical population gives rise to the non-classical population through this intermediate population in both mice and humans [11,12]. However, the ratio of monocytes in steady state conditions is different in mice and humans, with the former having almost equal numbers of the two populations and the classical subset predominating in the latter.

The frequency, activity, and fate of monocyte populations have been linked to many diseases, including autoimmunity, chronic inflammation, cardiovascular diseases, and cancer [13,14,15,16,17,18]. In cancer, classical monocytes are in many cases the source of tumor associated macrophages [19,20,21] and metastasis associated macrophages [22,23] that can promote primary tumor progression and metastases through a number of different mechanisms [14,24,25,26]. However, in some cases, such as in models of pancreatic cancer and glioblastoma, there can also be significant recruitment from yolk sac derived tissue resident macrophages [27,28]. The idea that intrinsic tumor features drive the tumor microenvironment landscape [29] has been reinforced by comparisons of the specific tumor types within a tissue, for example, by comparing metastatic brain tumors and primary brain tumors [28]. In mouse models of breast cancer, the origin of tumor and metastasis associated macrophages can be found in circulating monocytes [19,21] that are recruited via CCL2 synthesized at least in part by tumor cells [19,23].

There is also growing evidence that cancer has systemic influences on monocytes. For example, primary tumors exert systemic influences to generate the so-called pre-metastatic niche that enhances metastatic cell seeding at these distant sites [30,31,32]. The formation of this niche appears in part through effects in the BM and the subsequent mobilization of cells and their recruitment to tissue specific sites. These cells have been referred to as “immature” myeloid cells that are probably monocytes and their immediate post-differentiation stages that enhance metastatic cell trophism and seeding [32,33,34,35]. Several factors have been shown to be responsible for the formation of pre-metastatic niches, including fragments of extracellular matrix (ECM), various ECM modifying enzymes such as LOX1, growth factors such as VEGFA and tumor derived exosomes [30,32].

In several different human cancers, the ratio of classical to non-classical monocytes is also altered. For example, recent studies have shown dramatic increases in the non-classical to classical monocyte ratio in breast and endometrial cancer [36], and this increase of non-classical monocytes is negatively associated with tumor size and pathological stage in breast cancer [37]. In addition, monocytes isolated from patients with cancer have been shown to be transcriptionally altered [36,38,39]. These transcriptional alterations are profound enough to drive gene expression signatures that predict the presence of cancer [13,39]. However, whether similar changes occur in mouse models is unknown.

The PyMT transgenic mouse model (PyMT), in which the polyomavirus middle T protein is expressed under the control of the mouse mammary tumor virus promoter, whose expression is restricted to the mammary gland epithelium, is a well characterized model for studying ductal breast cancer. In this model, multifocal tumors develop spontaneously in the mammary glands and evolve in a manner that reflects stages of human ductal cancer [40,41]. Hence, this model has been used to study immunological perturbations that accompany mammary cancer progression. In this model, tumor progression leads to the dramatic expansion of circulating Gr1^+^ myeloid cells (mostly neutrophils) [42]. Focusing predominantly on neutrophils, Casbon et al. demonstrated an increase in the production of neutrophils within the BM and an expansion of monocytes within the spleen [42]. However, the use of the traditional anti-Gr1 antibody that detects both Ly6C and Ly6G antigens and thus both neutrophils (Ly6G^+^) and classical monocytes means that specific cell populations were not always assessed in this study. Nevertheless, these findings, combined with the findings in humans that circulating monocytes have altered population dynamics and distinct transcriptional signatures in cancer when compared to healthy controls, suggests important functions for monocytes that may be best interrogated in mouse models given the availability of genetic analysis in this species. Hence, we aimed to investigate the effect of tumor progression on the production, turnover, transcriptional signature, and composition of the monocyte compartment in the PyMT model.

## 2. Materials and Methods

### 2.1. Mice

Male B6Tg(MMTV-PyMT)^634Mul^/^Lellj^, a kind gift from Dr. Sandra J. Gendler (Mayo Clinic, AZ, USA), were bred with female WT or PyMT^−/−^ C57BL/6JCrl mice obtained from within the in-house existing colony. 8-week-old C57BL/6JCrl mice were purchased from Charles River Laboratories and used as as controls. FVBg(MMTV-PyMT)^634Mul^ were originally obtained from Dr William Muller and maintained in a colony in-house. Animals were housed and bred under standard conditions of care. All procedures involving mice were conducted in accordance with Arrive guidelines with licensed permission under the UK Animal Scientific Procedures Act (1986) and associated guidelines.

### 2.2. Flow Cytometry and Cell Sorting

Blood was collected by intra-venous (IV) bleed or cardiac puncture into a syringe containing 0.5 M EDTA (Fisher Scientific, Loughborough, Leicestershire, UK). Single-cell suspensions were prepared using red blood cell lysis buffer (Biolegend, Loughborough, Leicestershire, UK) for blood and red blood cell lysis buffer for BM (MERCK) and then via filtering with a 70µ filter. Spleens were Dounce homogenized and mononuclear cells obtained using LymphoprepTM (Stem Cell Technologies, San Diego, CA, USA). Single cell suspensions were blocked with CD16/32 (93) and stained with the following antibodies: MHCII (M5/114.15.2), CD45.2 (93), TREMl4 (16e5), CD11b (M1/70), CD115 (Afs98), CD3 (17A2), CD19 (6d5), Siglec-F (E50–2440), NK1.1 (PK136), Ly6C (hk1.4), Ly6G (1a8), CD11c (N418), CD135 (A2F10.1), Sca1 (D7), CD117 (2b8), CD127 (A7R34), TER119 (ter-119), and Streptavadin (BioLegend, Waterbeach, Cambridge, UK). The cells were resuspended with DAPI or propidium iodide (PI) solution (eBioscience, subsidiary of Fisher Scientific, Loughborough, Leicestershire, UK) and analyzed with a 6-Laser Fortessa cytometer (BD Biosciences, Wokingham, Berkshire, UK). For accurate cell number quantification, 123count eBeads™ Counting Beads (Thermo Fisher Scientifc, Cat No 01-1234-42) were used. The cells were sorted using FACS Aria Fusion (BD Biosciences). The flow cytometry data were analyzed using FlowJo (Three Star, Ashland OR).

For analysis, blood monocytes were defined as lineage^−^ (CD3, CD19, NK1.1), CD11b^+^, CD115^high^ and then assessed using both Ly6C and Treml4 to separate out Ly6C^high^ (Ly6C^high^/Treml4^−^), Ly6C^low^ (Ly6C^low^/TREML4^high^), and Ly6C^int^ populations (Appendix A). Blood neutrophils were defined as lineage^−^, CD11b^+^, CD115^low^, and Ly6G^+^ (Appendix A). BM monocytes were defined as lineage^−^ (CD3, CD19, NK1.1, Ter119, Ly6G), CD11b^+^, CD115^high^, and then assessed as Ly6C^high^ or Ly6C^low^ based on Ly6C expression alone (Appendix A). BM neutrophils were identified as Lineage^+^ CD11b^+^ cells.

For analysis and sorting of BM Ly6C^high^ monocytes, Macrophage Dendritic Progenitor cells (MDPs) and Common Myeloid Progenitors (cMoPs), live, CD115^high^ cells were selected as lineage^−^ (CD3, CD19, NK1.1, Ter119) and Ly6G^−^. Sca1^−^, CD127^−^ (IL7rα-) cells were gated into three populations based on CD117 (cKit) and CD135 (Flt3) expression, and each of these populations were assessed for Ly6C and CD11b expression: CD117^low^, CD135^−^, Ly6C^high^, CD11b^+^ monocytes, CD117^high^, CD135^−^, Ly6C^high^, CD11b^−^ cMoPs, CD117^high^, CD135^+^, Ly6C^−^, and CD11b^−^ MDPs (Appendix A). The sorted MDPs and cMoPs were cultured and picked, and cells were Giemsa stained to validate identity.

For the sorting of Lin^−^ Sca1^−^ c-Kit^+^ (LK), live cells were selected as lineage^−^ (CD3, CD19, NK1.1, Ter119) and Ly6G^−^. LK cells were gated as Ly6C^−^, CD11b^−^, CD117^+^, and Sca1^−^ (Appendix A). The observation of all types of colonies in CFU assays from these cells confirmed their multipotency, particularly the presence of red tinged BFU-E colonies (data not shown).

For the analysis of primitive BM progenitors, myeloid progenitors were defined as lineage^−^ (CD3, CD19, NK1.1, CD11b, Ly6G, Ly6C, Ter119) and CD127^−^ and then divided into LK and LSK as CD117^high^, and Sca1^−^ and Sca1^+^ respectively. The LSK population was divided into CD135^+^ multipotent progenitors (Flt3^+^ MPPs) and CD135^−^ hematopoietic stem progenitor cells (HSPCs). All CD135^−^ cells were then divided into CD48^+^ cells, representative of the CD135^−^ MPPs (Flt3^−^ MPPs), and CD48^−^ and CD150^+^ hematopoietic stem cells (HSCs). The LK population was divided into megakaryocyte-erythrocyte progenitors (MEPs) (CD34^−^, CD16/32), CMPs (CD34^+^, CD16/32^−^), and GMPs (CD34^+^, CD16/32) (Appendix A).

For the spleen, following the selection of live single cell populations, lineage^−^ (CD3, CD19, NK1.1, Ter119, Ly6G) and CD115^high^ cells were gated and then divided according to expression of CD135 and CD117, with the monocytes defined as double negative and MDPs as double positive. Monocytes were gated as CD11b^+^, Ly6C^high^ and MDPs as CD11b^−^, Ly6C^−^ (Appendix A).

### 2.3. CFU Assays

Progenitor cells were sorted into 1.5 mL Eppendorf tubes containing Iscove’s modified Dulbecco’s medium (IMDM) medium, and cells were regained by centrifugation at 400× *g_av_* and re-suspended in IMDM medium at a concentration of 600 cells/300 µL for MDPs or 800 cells/300 µL for LK cells. 300 µL of cell suspension was added to 3 mL aliquots of MethocultTM M3534 (STEMCELL Technologies, Waterbeach, Cambridge, UK,) in Falcon™ Round-Bottom Polystyrene Tubes (Falcon, Seaton Delaval, Tyne and Wear, UK), vortexed and plated in 35 mm petri dishes with duplicates for each sample. Plates were incubated at 37 °C 5% *v*/*v* CO_2_ ≥ 95% humidity in a 15 cm dish with an additional petri containing Dulbecco’s PBS (DPBS). Colonies were counted at 12–14 days. For morphology, colonies were picked and cytospun (Shandon Cytospin II) and then stained with rapid Romanowsky stain solutions A, B, C, (TCS Biosciences Ltd., Botolph Claydon, Buckingham, UK).

### 2.4. Determination of Cells in DNA Synthesis

Mice were injected intraperitoneally with 100 mL BrdU in DPBS (10 mg/mL; Sigma Aldrich, subsidiary of Merck Life Science UK Limited, Gillingham, Dorset, UK). Single cell suspensions were prepared and incubated with primary antibodies as specified. Live/dead stain was applied (Zombie Aqua^TM^ Fixable Viability Kit Biolegend, Fixable Viability Dye eFluor™ 780 eBioscience). Cells were fixed and washed with FoxP3-staining-buffer-set (eBioscience) and then incubated with DNase at 37 °C for 30 min (1 mg/mL in D-PBS, Sigma Aldrich, DNase-solution 30 mL DNase-stock + 960 mL D-PBS + 10 mL MgCl_2_). Cells were washed and stained with anti-BrdU-antibody (Alexa Fluor^®^ 488 anti-BrdU Antibody, EBioScience) for 30 min at RT. For each group and in all experiments, background BrdU levels were set using a non-DNase treated control. Representative staining and gates are shown in Appendix A.

### 2.5. RNA-Seq of Monocytes

For scRNAseq, the total population of monocytes (defined as lineage^−^ (CD3, CD19, NK1.1, CD11b^+^, CD115^high^) were sorted into low binding, conical 96 well non-skirted plates (Sigma) containing 2 µL of lysis buffer (1.9 mL of 0.2% *v*/*v* Triton-X l00 *v*/*v* + 0.1 mL of RNasin Plus RNase inhibitor (10,000 U/mL, Promega)). Smart-seq2 protocol was performed as described in Picelli et al. [43]. Libraries were pooled in a 1:1 ratio and sequenced on one lane of an Illumina HiSeq2500. Raw reads were aligned onto mouse mm10 assembly with BWA. Only uniquely aligned reads were retained and quantified using htseq [44]. Reads were normalized using RSEM [45]. Clustering and differential gene expression analysis was done using Seurat package in R [46].

For deep RNA sequencing, monocytes were sorted as detailed above and RNA was isolated using 475 µL TRIzol™ LS Reagent (Sigma Aldrich, subsidiary of Merck Life Science UK Limited, Gillingham, Dorset, UK). Poly A enriched mRNA was fragmented in 2x Superscript III first-strand buffer with 10 mM DTT (Invitrogen, Carlsbad, CA 92008, USA) via incubation at 94 °C for 10 min and then immediately chilled on ice. Samples of 10 μL of fragmented mRNA, 0.5 μL of random primer (Invitrogen), 0.5 μL of Oligo dT primer (Invitrogen), 0.5 μL of SUPERase-In (Ambion), 1 μL of dNTPs (10 mM), and 1 μL of DTT (10 mM) were heated at 50 °C for one minute. At the end of incubation, 6 μL of water, 1 μL of DTT (100 mM), 0.1 μL actinomycin D (2 μg/μL), and 0.5 μL of Superscript III (Invitrogen) were added and incubated in a PCR machine using the following conditions: 25 °C for 10 min, 50 °C for 50 min, and a 4 °C hold. The product was then purified with Agentcourt RNAClean XP beads (Beckman Coulter) according to the manufacturer’s instruction and eluted with 10 μL nuclease-free water. The RNA/cDNA double-stranded hybrid was added to 1.5 μL of blue buffer (Enzymatics), 1.1 μL of dUTP mix (10 mM dATP, dCTP, dGTP, and 20 mM dUTP), 0.2 μL of RNase H (5 U/μL), 1.2 μL of water, and 1 μL of DNA polymerase I (Enzymatics). The mixture was incubated at 16 °C for 2 h. The purified dsDNA underwent end repair using blunting, A-tailing, and adaptor ligation barcoded adapters (NextFlex, Bio Scientific). Libraries were PCR-amplified for 9–14 cycles, size selected by gel extraction, quantified using a Qubit dsDNA HS Assay Kit (Thermo Fisher Scientific) and sequenced on a NextSeq 500.

Data was mapped to custom genomes using STAR [47] with default parameters. All samples were assessed for quality using the FastQC (Babraham Bioinformatics, 2010) package with MutliQC [48]. Unique mapping rates and read depths were checked with a cut-off of 90% minimum uniquely mapped reads and >20 × 10^6^ total read depths unless otherwise specified. Additionally, correlation between samples and clonality were checked across all samples, and all samples were visually inspected on the UCSC genome browser. Adapters were trimmed and counts were generated using HOMER, available at http://homer.ucsd.edu/homer, accessed 27 June 2019. Differential gene expression (DGE) was assessed with the DESeq2 package with the false discovery rate < −0.05 and betaPrior = TRUE. Data visualization was undertaken in R studio. For average expression and log fold change (LFC) visualization, LFC shrinkage using the alpegm method was used. Pathway analysis was undertaken using Metascape [49] (http://metascape.org/gp/index.html, accessed 27 June 2019). To convert mouse genes to human orthologues the R package g:Orth [50] was used with the following command line: orth=gorth(rownames(mousedf), source_organism = “mmusculus”, target_organism = “hsapiens”, region_query = F, numeric_ns = ““, mthreshold = 1, filter_na = T, df = T). As there were multiple orthologue matches, the genes with the highest variance were selected.

### 2.6. Motif Enrichment Analysis

Motif enrichment analysis was performed on HOMER findMotifsGenome.pl with parameters “-size 200-mask” to identify de novo motifs and their matched known motifs [51]. Motif analysis was performed using the promoter sequences (−500 to +50 bp from the transcriptional start site) of the identified 393 down-regulated genes of Ly6cHi cells and the 108 down-regulated genes of Ly6cHi cells from the bone marrow. The background sequences were the default GC-matched random genomic background.

### 2.7. Data Access

Single Cell datasets are deposited in GEO: P02E01-GSE183838, P02E02-GSE184870, DEG in Appendix A.

### 2.8. Statistical Analysis

All quantitative analyses were based on at least three sample replicates. Data are presented as means ± SD using GraphPad Prism. Independent-sample student *t* test were performed (SPSS). NS, not significant; * *p* < 0.05; ** *p* < 0.01; *** *p* < 0.001.

## 3. Results

### 3.1. Tumor Bearing MMTV-PyMT Mice Show Increased Numbers of Monocytes

The effects of cancer on monocytes and neutrophils were determined in the blood, BM, and spleen using the polyoma middle T mouse model of breast cancer on a BL6 genetic background. The diagnostic flow cytometry panels defined in the materials and methods were used together with counting beads to calculate the frequency of specific cell populations in the blood (Figure 1A,B). To perform this study, control and cancer bearing mice were bled bi-weekly from 8 weeks of age until the age that the tumors were 24 mm diameter corresponding to the end stage according to our approved animal protocol. As tumor onset is sporadic on the BL6 background, we grouped the tumor bearing mice into those that had early and late stage carcinomas as defined histologically [40]. There was a 1.4-fold increase in the total cellularity of the blood in mice with tumors (Figure 1C). This expansion was limited to the CD11b^+^ fraction, with no expansion in other circulating CD11b^−^ cells, such as T or B cells (Figure 1A,C). Using CD115 and Ly6G expression to distinguish monocytes (CD115^high^) from neutrophils (Ly6G^+^), we found that both monocytes and neutrophils increased in numbers even in control mice with age, but this effect was greater in late cancers and accounted for most if not all of the expansion of CD11b^+^ cells (Figure 1D,E). However, the expansion of monocytes was more modest than observed for neutrophils (2.1 and 3.6-fold, respectively) and was restricted to mice bearing late-stage tumors, while the increase in neutrophils was significant even in early cancers (Figure 1E,F). Notably, the distribution of monocytic populations was unaltered, with equivalent percentages of Ly6C^high^, Ly6C^int^, and Ly6C^low^ monocytes in age-matched non-tumor control and tumor-bearing mice (Figure 1G). Consistent with this unchanged ratio, there was an increase in the total number of each monocytic population, with the increase in the classical population being the highest (Appendix A). Analysis of the BM from femurs also showed an expansion in monocytes and neutrophils (Figure 1H), suggesting that cancer affects myelopoiesis in this tissue and that this might contribute to blood monocytosis and neutrophilia.

### 3.2. Tumor-Bearing Mice Exhibit Elevated Proliferation of BM Monocytes

The purpose of this study was to study monocyte biology, as neutrophil biology had been studied in detail before [42]. Thus, to examine the underlying cause of cancer-related BM and blood monocytosis, the frequency and differentiation potential of monocyte progenitors in the BM was assessed. In the B6 PyMT model, there was no increase in the frequency of either MDPs or cMoPs in the BM of tumor-bearing mice. Furthermore, there was no change in the MDP upstream progenitor defined by lin^−^, cKit^+^, Sca1^−^ cells (LK) (Figure 2A). To support this conclusion, there was no difference in the ratio of colonies formed when LK cells were cultured in vitro in standard colony forming unit assays (Figure 2B), nor any increase in the potential of LK, MDPs or cMoPs from mice bearing tumors to form colonies in this assay (Figure 2C). These data suggest that both the fate potential and proliferative capacity of monocyte progenitors in the BM remain unaltered in tumor-bearing mice.

To determine if the proliferation of these progenitors in vivo was altered by presence of cancer, we identified cells in S-phase by intraperitoneal injection of the thymidine analogue, BrdU, and analyzing nuclear incorporation after 1 h. Supporting the CFU assays, there was no difference in the incorporation of BrdU between cancer and control in the Lin^−^, Sca1^+^, c-Kit^+^ (LSK), LK, GMPs, MDPs or the cMoPs (Figure 2D). Monocytes can also proliferate in the BM in normal and cancer bearing mice [11,19] and therefore their frequency in S-phase in vivo was also measured in the same 1 hr BrdU pulse experiment. Strikingly, mice bearing late-stage tumors exhibited significantly greater proliferation of BM monocytes, while proliferation of BM neutrophils and CD11b^−^ cells was equivalent to that found in control animals (Figure 2E).

The increased blood monocyte population in the blood could be due to either this increased proliferation in the BM and/or to their proliferation, survival and clearance once they have entered the blood. To determine the latter, pulse-labelling with BrdU was used in mice with late-stage tumors or age-matched controls to determine the half-life of blood monocytes over a 10-day period as described by [11]. No BrdU^+^ monocytes were detected in the blood at 1 h post-BrdU pulse confirming that proliferation of blood monocytes neither occurs in control nor cancer-bearing mice (Figure 2F). Peak labelling with BrdU occurred at 24, 72, and 96 hrs for Ly6C^high^, Ly6C^int^ and Ly6C^low^ monocytes respectively (Figure 2F), consistent with previous estimates of the sequential differentiations and different half-lives of these populations [11]. However, at no timepoint was there a difference in the level of BrdU^+^ cells apparent in the monocyte subsets from control and cancer-bearing mice (Figure 2F). Together these findings strongly suggest that the increase in blood monocytes found in tumor-bearing mice arises from elevated proliferation of Ly6C^high^ monocytes in the BM while the survival, half-life and differentiation of these cells subsequently remains unaltered following their entry into the blood.

The spleen contains a significant reservoir of Ly6C^high^ monocytes during homeostasis [52] but may also be a site of significant extramedullary monopoiesis under conditions of stress [53,54]. Notably, mice with late-stage cancer exhibited splenomegaly (Figure 3A). While there was an increase in the number of mononuclear cells (MCs) in tumor bearing mice versus age-matched controls (T cells, B cells and Monocytes isolate from gradient separation) (Figure 3B), the density of mononuclear cells per gram of spleen was equivalent in age-matched controls versus tumor bearing mice (Figure 3C). However, splenic Ly6C^high^ monocytes formed a greater proportion of live cells and their estimated frequency was also significantly increased in tumor bearing versus age-matched controls (Figure 3D,E). To determine if monopoiesis in the spleen contributes towards the increase in circulating monocytes observed, we looked for evidence of splenic MDPs by flow cytometry and local proliferation of monocytes at 1 h post BrdU injection. However, the frequency of splenic Ly6C^high^ monocytes that incorporated BrdU was low (1–2% BrdU^+^) in both control and tumor-bearing mice, despite proliferation of other mononuclear cells (Figure 3F), and splenic MDPs could not be detected in control mice or tumor bearing mice (data not shown). These data suggest recruitment to or retention within the spleen from the blood is the predominant cause of the elevated monocyte count.

### 3.3. Monocytes from the Blood of Mice with Tumors Are Transcriptionally Altered

Having observed a difference in the proliferation of BM monocytes in tumor-bearing mice, we then examined whether cancer led to a transcriptional shift in the monocytes. First, to establish if there were transcriptionally distinct subpopulations of monocytes that differed in tumor-bearing mice, single cell sequencing was performed using SMARTseq on the total monocyte population, as described in the Materials and Methods section. Plotting the data using a uniform manifold approximation and projection (UMAP) identified two major populations and one minor one (Figure 4(Ai)) and indicated that there were no new populations of monocytes occurring in response to cancer (Figure 4(Aii)). Marker analysis showed the two major populations corresponded to classical (Ly6C^+^, CCR2^+^) and non-classical monocyte (Treml4^+^, NR4A1^+^) populations, respectively (Figure 4(Aiii)). As classical monocytes differentiate into non-classical ones, the second small population is likely the intermediate monocyte population [11]. This contention is supported by the marker analysis (Figure 4(Aiii)), as the two defining markers for each major population are represented in this intermediate one.

Having established that monocyte populations in cancer were similar to those in control mice, we sorted monocytes according to Ly6C expression and undertook deep RNA sequencing. Principle component analysis (PCA) revealed monocytes to primarily cluster according to monocyte subset rather than condition (Fig 4B, Appendix A). However, separate analysis of Ly6C^high^ and Ly6C^low^ populations revealed that while there was very little alteration in the Ly6C^low^ monocytes in tumor-bearing mice (with just 7 DEGs revealed) (Appendix A), the Ly6C^high^ monocytes were altered, clustering separately from those of the control mice (Figure 4C). Ly6C^high^ monocytes featured a total of 611 differentially expressed genes (DEGs) (adj *p* value < 0.05) between the cancer and control mice (Figure 4C; Appendix A). There was an almost two-fold difference in the number of genes that were down-regulated versus those that were up-regulated (393 down, 218 up). To identify potentially important genes, the chemokines and transmembrane receptors relevant to monocyte biology were explored along with transcription factors (Figure 5A). Key genes modulated by IFNs were downregulated. This included the type II IFN induced chemokine gene, *Cxcl10*, and genes for MHCI and MHCII proteins. The lineage determining transcription factors (LDTFs), *Fosb* and *Jun*, were down-regulated. Signal transducer and activator of transcription (STATs) involved in IFN response, *Stat1*, and *Stat2* were down-regulated along with the regulatory TFs *Socs1, Socs3, Socs6*, *Jak2*, and *Jak3*. To further explore whether the DEGs may be functionally related, genes were analyzed for pathway enrichment using Metascape [49]. In the down-regulated list of genes, pathways were highly enriched (log q.value > 10) in relation to IFN signaling and antigen presentation (Figure 5B).

To investigate transcription factors associated with down-regulated genes identified in Figure 4C, we performed de novo motif enrichment analysis of their corresponding promoters. This analysis identified a motif for STAT1:STAT2 as the most highly enriched motif (Figure 5C), providing evidence that the down-regulation of the mRNAs encoding STAT1 and STAT2 is functionally related to genes exhibiting reduced expression in the Ly6C monocytes of tumor-bearing mice. In contrast, motifs recognized by FOSB or JUN were not highly enriched. This could reflect the functional redundancy of the FOS/JUN family of transcription factors or the possibility that they primarily function at distal regulatory elements. The remaining highly enriched motifs (ELF, BACH, ZNF384, and RUNX) correspond to binding sites for factors that typically exhibit high enrichment at promoters.

### 3.4. Transcriptional Changes Occur within the Bone Marrow

The above data indicates there are transcriptional alterations to circulating monocytes in the context of cancer. Hence, we wanted to ascertain at what stage of their development this conditioning may be taking place. For this purpose, the dominant Ly6C^high^ monocytic population representing ~90% of these cells from the BM were sequenced. PCA revealed that the cancer samples did not cluster is the same way as the controls (Figure 6A). Further analysis revealed 158 genes to be differentially expressed between the cancer and control samples (50 up- and 108 down-regulated in cancer with a adj *p* value < 0.05) (Figure 6B, Appendix A). Even though there were fewer genes altered in BM monocytes, just under half of those genes that are down-regulated in BM monocytes by cancer were also down-regulated in Ly6C^high^ blood monocytes (Figure 6C; Appendix A. These 47 common down-regulated genes were highly enriched for type I IFN pathways (Figure 6D).

### 3.5. Transcriptional Shifts in the Mouse PyMT Model Are Not Orthologous to Patients with Breast Cancer

Having identified the DEGs in cancer and control mice and the relevant pathways that were altered, we then compared these to the human data on total circulating monocytes (classical and non-classical) in patients with breast cancer and healthy controls from a previously published cohort [13]. This reference also gives full details of the patient and control cohorts used in the human study. To undertake this comparison, a list of 865 DEGs between the breast cancer patient and control healthy volunteer samples was used (selected using a criterion of absolute LFC > 1.5) from this human data set [13]. Comparing these datasets, there were very few DEGs common to both BL6 mice and humans (Figure 7A,B). In case the strain of the mouse affected the findings, circulating monocytes were also sequenced from FVB mice with PyMT tumors and age-matched controls. This generated a third list of DEGs with 83 up-regulated and 163 down-regulated between strains (see Appendix A). However, again there were very few DEGs common to mice and humans (Figure 7A–C). The lineage determining transcription factor, *Jun*, was the only gene that was commonly down-regulated in all the DEG lists. To identify if there were common pathways affected, despite a lack in shared DEGs, joint pathways analysis was undertaken. This revealed some commonly enriched down-regulated pathways, particularly with respect to metabolism, cell differentiation, survival, and migration. However, the main pathway that had been identified as down-regulated in classical monocytes in mice, IFN signaling, was up-regulated in human cancer monocytes (Figure 7D). These data suggest different responses to cancer in the two species. These two mouse strains were chosen as opposite ends of a cancer-susceptibility spectrum [55]. BL6 represents a strain that is tumor resistant and the other, FVB, is cancer susceptible, as shown by the time of onset and transition to late-stage malignancy of the tumors (Figure 7E,F). Strikingly, although there were common DEGs between strains (Figure 7A,B; Appendix A), there were more DEG that were different, particularly apparent in the FVB strain (Figure 7A,B). Pathway analysis, however, suggested that there are many similar pathways affected. The down-regulation of IFN signaling, in particular, was seen in both strains, although to a lesser extent in FVB. In addition, there were more up-regulated pathways in FVB (Figure 7D).

## 4. Discussion

It is well known that cancer causes systemic effects, a process particularly well documented by the induction of cachexia [56]. This metabolic disease tends to occur during the advanced stages of the disease. However, data is accumulating on mice and humans which shows that cancer also causes systemic effects during the early stages of the disease. An example of this phenomena is the establishment by early tumors of the “pre-metastatic niche’ that biases the homing of metastatic cells and enhances their seeding [57,58]. The formation of this niche depends on the secretion of ECM components such as fibronectin and exosomes as well as the mobilization of BM derived cells, such as monocytes and neutrophils, to these sites [18]. The earliest description of what some have described as sterile inflammation was the accumulation of CD11b+ VEGFR1+ myeloid cells [59], which are most likely derived from classical monocytes that undergo differentiation in response to the “niche” environment [35,60]. This pre-metastatic niche concept has yet to be established in humans, but there are increasing numbers of reports that cancer alters monocyte biology. This alteration is manifested by the enhancement of monocyte numbers [61], an increase in the non-classical monocyte population [13,37], and alterations in total monocyte transcriptomes [13,38,39]. Transcriptional signatures from monocytes were derived from breast and colorectal cancers that were able to predict the presence of cancer [13,39]. However, these signatures did not overlap, possibly due to the different cancers or to different transcriptional profiling methodologies.

In this study, we hypothesized that mouse models of breast cancer would have similar impacts on monocytes as those found in humans and thereby allow experimental manipulation to determine their mechanism. In this study, we chose the well-characterized PyMT model, which has reliable cancer proregression, is, importantly, metastatic, and is similar to human luminal cancer [40]. However, whilst we documented a significant monopoiesis in this model, there were no changes in non-classical to classical monocyte ratios in the blood, nor any indications of new populations by FACS or single cell RNAseq. This failure to see ratio changes in mice compared with humans may simply reflect the starting populations of classical and non-classical monocytes, which in humans are approximately 9 to 1, respectively, but in mice are approximately equal. Furthermore, there were no other significant monocytic populations which are found in the blood of human cancers [62], such as the myeloid derived suppressor cells, indicated by the analysis of side-scatter, nor was there an emergence of other monocytic subtypes, such as SatM monocytes, found during fibrosis [63]. The failure to find such populations might be related to the sorting procedure in mice, which was directed by traditional monocyte markers.

Our data also indicated that the monocytopoiesis induced by cancer was not due to an increase in BM progenitors nor to the changed half-lives of circulating monocytes, but that it was due to the enhanced proliferation of monocytes in the BM. Others have shown the spleen to be a source of monocyte progenitors in lung cancer models [64] as a result of the angiotensin II produced by cancer remotely increasing the frequency of the HSC that migrate to the spleen [65]. However, although the number of splenic monocytes increased along with the splenomegaly, we found no evidence that cancer increased their proliferation nor enhanced the very low frequency of progenitors in the spleen in this model of breast cancer.

The elevation of CSF1 is the most obvious explanation for the increased proliferation of monocytes in the BM. Elevated levels of circulating CSF1, a proliferation and survival factor for monocytes and macrophages, have been reported at late stages of cancer in the spontaneous FVB PyMT model of breast cancer [42] as well as in human cancers [66,67]. In the PYMT mouse model, this elevation was preceded by elevated G-CSF (*Csf3*) and an earlier expansion of neutrophils when compared to monocytes, corresponding to our findings regarding cell population dynamics [42]. Similarly, in the circulation, the lack of changes in the half-lives of the increased numbers of monocytes in cancer suggest the presence of sufficient CSF1 to maintain their viability. In the BM, interferon signaling has also been shown to inhibit the proliferation of differentiated hematopoietic cells, but not in the case of HSC proliferation [68]. Thus, the down-regulation of this signaling pathway, as observed in our sequencing data, combined with elevated CSF1, could explain the relatively specific effect of tumors on the BM proliferation of monocytes.

Transcriptomic analysis indicated relatively small changes in the classical monocyte population in response to cancer, with the variance between monocyte sub-sets having the greatest variance. The only available human dataset that was used for comparison was collected using the entire monocytic population. The proportion of non-classical monocytes in patients with breast cancer has been shown to be higher than in healthy controls [13] and this may have masked the more subtle effects of cancer on the classical population. However, the translation of these findings in mice to humans was further limited by the lack of orthologous genes in our datasets. Single cell RNAseq data comparing lung tissue isolated from murine KP1.9 lung adenocarcinoma and seven human primary NSCLC samples suggest that there should be homologous genes between human and mice [8]. However, the lack of depth of sequencing in this study led to comparisons of common monocytic genes that define lineages. This data was true in our study but we concentrated only on the DEGs that could be identified by the depth of sequencing. These genes showed a lack of concordance between species.

In mouse models, the immune profile of the tumor microenvironment is known to be specific to the genetic mutations within the tumor, and the same is thought to apply to human cancers [29]. The systemic influences of cancer are also dependent on specific gene mutations [29]. The PyMT model is beneficial because of robust cancer development with histology that mirrors human ductal adenocarcinoma and metastatic capacity. However, this model is driven by the expression of a viral oncoprotein [40]; thus, the mutational profile does not reflect those found in human breast tumors. Furthermore, even when the cancer is driven by the same oncoprotein, there are significant differences in gene expression according to strain (FVB vs. BL6). Since timing of onset, age of transition, and metastatic capacity in these two strains is very different, this suggest that genetics even within inbred species can profoundly influence systemic response to cancer. These changes maybe coincidental or causal, and if the latter, may therefore open genetic analysis as performed for quantitative trait loci for metastasis between these strains [69]. In addition, it should be appreciated that the development of mouse mammary cancer in the PyMT model is over a few weeks, while in humans it can be over many years. These timing differences limit exact comparisons between the species but do suggest this PyMT mouse model maybe inadequate for exploring mechanisms behind systemic changes seen in humans. It will remain to be determined in other metastatic mouse cancer models whether this is true.

Nonetheless, the findings in mice are still interesting. The proliferation of Ly6C^hi^ BM monocytes leads to an elevated number of blood monocytes that then circulate with normal half-life. Similar increases in monocyte numbers have been described in human cancers. The comparison of differentially expressed genes in the BM samples, with the larger numbers of genes differentially expressed in the blood compared to BM, suggest that the systemic influence of cancer begins in the bone marrow, particularly interferon signaling, but that the major effects occur in the circulation. These data indicate that an organism is sensing the presence of cancer perhaps through the immune detection of neo-antigens, and/or through the cancer production of cytokines, exosomes, or debris. Indeed, in the context of chemotherapy stress, tumors produce exosomes that modify monocytes so that they are preferentially recruited to metastatic sites where they promote tumor cell extravasation [60]. Overall, our data suggests that the cancer education is a two-step process whereby cancer affects the transcription and proliferation of Ly6C^hi^ monocytes in the bone, and, once the monocytes enter the circulation, further and more significant transcriptional differences are overlaid upon pre-existing changes.

## 5. Conclusions

There is a systemic impact on circulating and bone marrow monocytes in a mouse model of breast cancer. This is characterized by monopoiesis and a splenomegaly. The monopoiesis is due to enhanced monocyte cell proliferation in the bone marrow without effects on progenitor cell frequency. In the circulation, single cell sequencing indicates that there are no new monocyte populations in cancer and that their differentiation and turnover is unaffected.

Cancer also induces systemic effects on the transcriptome of monocytes. This is first evident in the bone marrow but more pronounced in the circulation. The down-regulation of interferon signaling is the major pathway affected. The genetic background differences between mouse strains that give cancer prone or resistant phenotypes and carry the same oncoprotein differentially alter monocyte transcriptional profiles. The comparison of mouse monocyte transcriptional changes with those found in human breast cancer patients also showed no similarity. These data indicate that cancer causes systemic effects on circulating monocytes which are strongly influenced by genetic background.

## Figures and Tables

**Figure 1 cancers-14-00833-f001:**
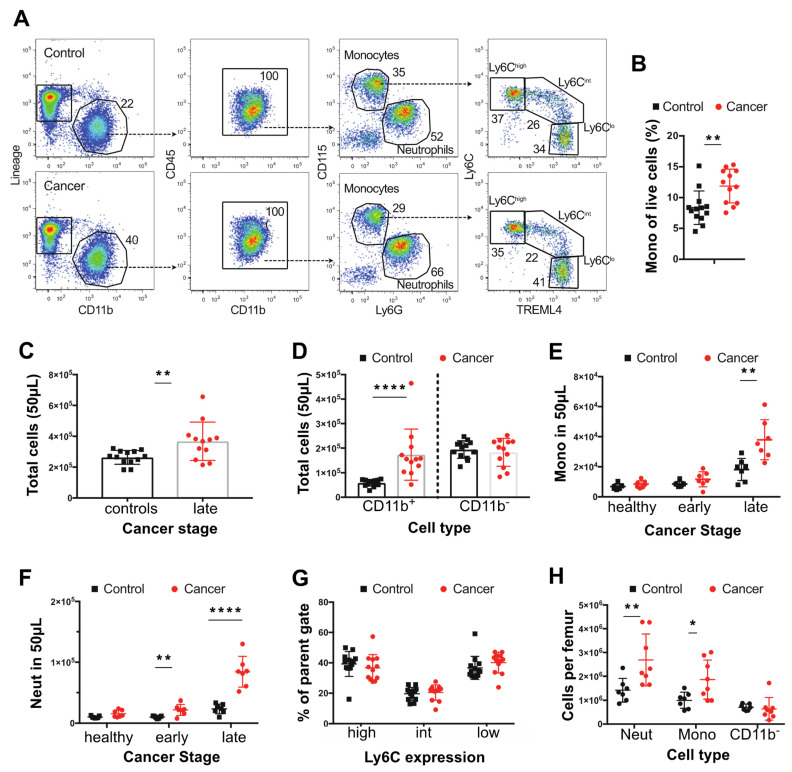
Changes in blood and bone composition of immune cells in PyMT tumor-bearing mice. (**A**) Representative dot plots showing identification of blood monocytes in control mice and mice with ~20 mm tumor (late) late-cancer. (**B**) Proportion of live blood leukocytes that are CD115^+^ CD11b^+^ monocytes in mice with late-stage cancer versus controls. (**C**) Total cells in 50 μL of blood in mice with ~20 mm tumor (late) late-cancer (red dots *N* = 12) versus controls (black dots *N* = 13). (**D**) Total number of CD11b+ and CD11b- cells in mice with late-stage cancer versus controls. (**E**) Total circulating monocytes prior to tumor development (healthy), at onset of tumors (early), and at late stage versus control WT mice. (**F**) Total neutrophils (Neut) prior to tumor development (healthy), at onset of tumors (early), and at late stage of cancer. (**G**) Percentage (%) of monocytes subpopulations at late stage compared with control WT mice. (**H**) Neutrophils, monocytes, and lineage+ CD11b- cells per femur in mice with late stage cancer versus controls. Cancer samples are shown in red and age-matched controls in black. * *p* value < 0.05, ** *p* value < 0.01, **** *p* value < 0.0001, multiple *t*-test. Experiments were conducted in duplicates with littermate and co-housed groups of *n* = 6 (3 PyMT + ve) and n = 8 (4 PyMT + ve) in each replicate for the blood and co-housed groups of *n* = 6 (3 PyMT + ve) and n = 9 (5 PyMT + ve) for the bone.

**Figure 2 cancers-14-00833-f002:**
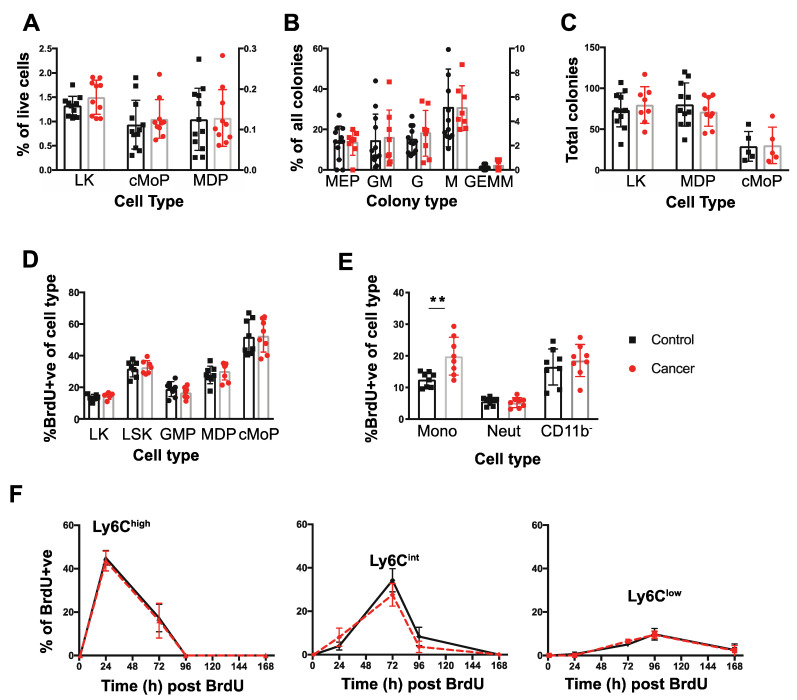
Dynamics of monopoiesis in tumor bearing mice. (**A**) Proportion of LK, cMoP, and MDP cells within the BM shown as a percentage (%) of total live cells extracted from the bone marrow. (**B**) Distribution of colony types within the LK CFU methylcellulose assays expressed as a % of each colony type from the total colony number. (**C**) Isolated LK cells, MDPs, and cMoPs progenitors were cultured in CFU assays in methocellulose for 12–14 days and the total number of colonies present were quantified. (**D**) Nuclear BrdU incorporation in the BM progenitors (LK, LSK, GMP, MDPs, cMoPs) 1 h post injection in mice with late cancer or age-matched controls. (**E**) BrdU nuclear incorporation in BM monocytes, neutrophils (Neut) and lineage+ CD11bcells 1 hr post injection in mice with late cancer or age-matched controls. (**F**) Level of BrdU+ circulating monocytes within each population over time. From left to right: Ly6Chigh Ly6Cint, Ly6Clow. Late cancer samples are shown in red and age-matched controls in black. ** *p* value < 0.01, unpaired *t*-test. For LK cells, sorts were conducted on 3 days in triplicates of co-housed littermate groups of *n* = 4 (two cancer), *n* = 11 (four cancer), and *n* = 5 (two cancer). For MDPs, sorts were conducted on 3 days with co-housed littermate groups of *n* = 6 (three cancer), *n* = 4 (two cancer), and *n* = 10 (five cancer). For cMoPs, sorts were conducted on 1 day with *n* = 10 (five cancer). For the BrdU, BM experiments were conducted in triplicates of co-housed littermate groups of *n* = 6 (three cancer), *n* = 6 (three cancer), and *n* = 4 (two cancer). For the BrdU blood tracing, experiments were conducted in duplicates of *n* = 6 (three cancer, three controls/group).

**Figure 3 cancers-14-00833-f003:**
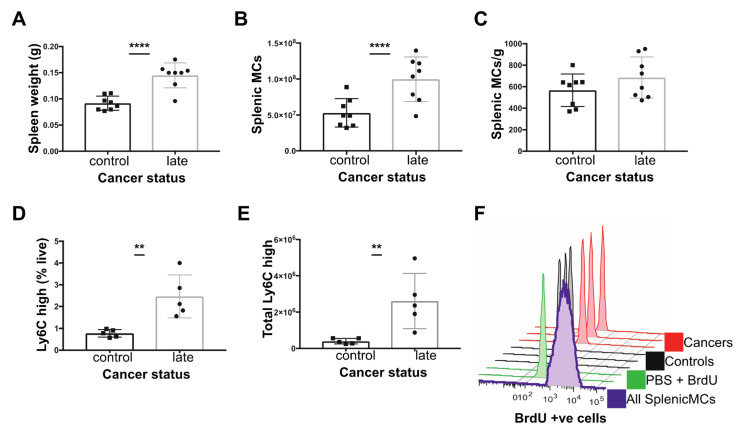
Characterisation of the spleen in C57BL/6 mice with late cancer. (**A**) Splenic weight. (**B**) Total splenic mononuclear cells (MCs; T cells, B cells, monocytes) (**C**) Total splenic MCs (x106) normalised to splenic weight. (**D**) Ly6Chigh monocytes as % of live cells. (**E**) Total Ly6Chigh monocytes in spleen. (**F**) Histogram of splenic BrdU levels in: all Splenic MCs in spleen (purple), PBS with no DNase control (green), Ly6Chigh Mo from controls (n = 3, black) and Ly6Chigh monocytes from late cancer mice (n = 3, red) 1 h post injection of BrdU. ** *p* value < 0.01, **** *p* value < 0.0001, unpaired *t*-test. Experiments were conducted in duplicates of co-housed littermate groups of n = 6 (3 cancer) and n = 4 (2 cancer).

**Figure 4 cancers-14-00833-f004:**
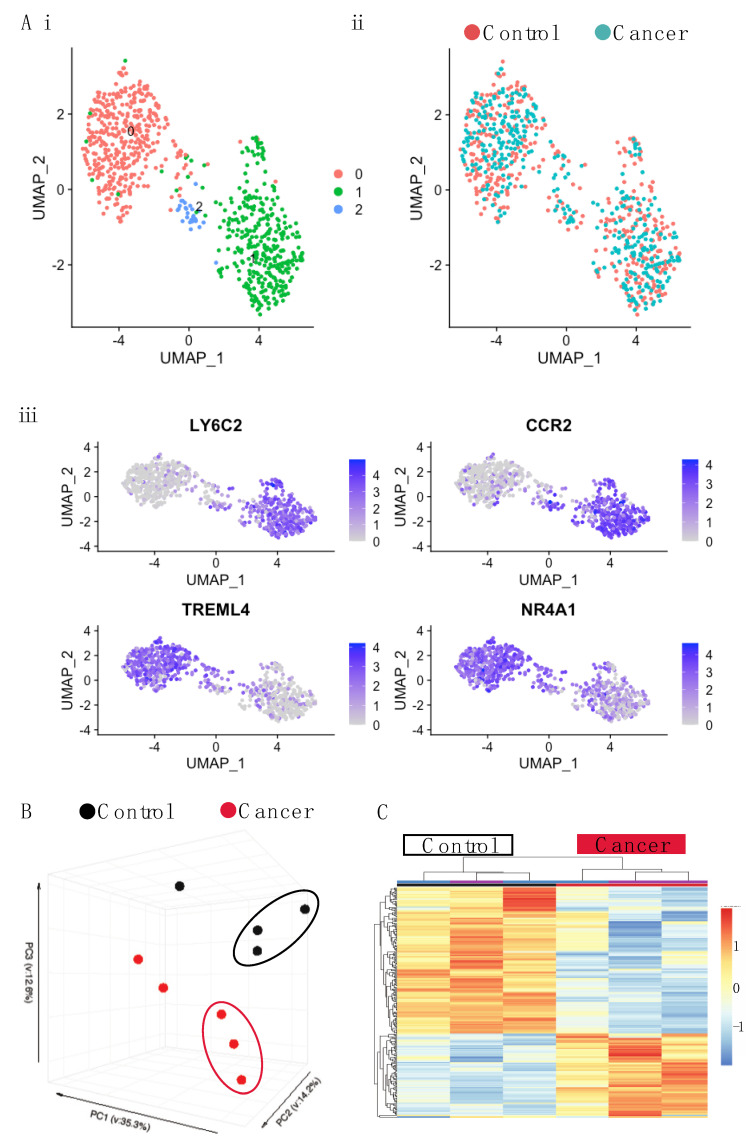
Single cell and bulk RNA-seq of blood monocytes in C57BL/6 mice with late stage cancer. (**A**) UMAP plot for single cell sequencing (SCseq) of total monocytes cells from control and cancer-bearing mice. (**i**) Distribution of all monocytes combined shows three populations: pink, blue, and green. (**ii**) Distribution of control monocytes (pink) and tumor (blue). (**iii**) Marker analysis showing classical (LY6C2, CCR2) and non-classical (TREML4, NR4A1) monocytes. (**B**) PCA of gene expression derived from bulk RNAseq of Ly6Chigh Monocytes. (**C**) Gene expression heatmap of Ly6Chigh blood monocytes differentially expressed genes (DEGs) between control and cancer samples with a q.value < 0.05. Samples are arranged horizontally. Sample characteristics are provided in horizontal bars for each column, denoting cancer in red and control in black and the date on which samples were sorted in purple and blue. Genes are arranged vertically; within the heatmap, red indicates up-regulation and blue indicates down regulation based on the tpm z-score (range −2, 2). Samples are clustered using complete linkage and Pearson correlation.

**Figure 5 cancers-14-00833-f005:**
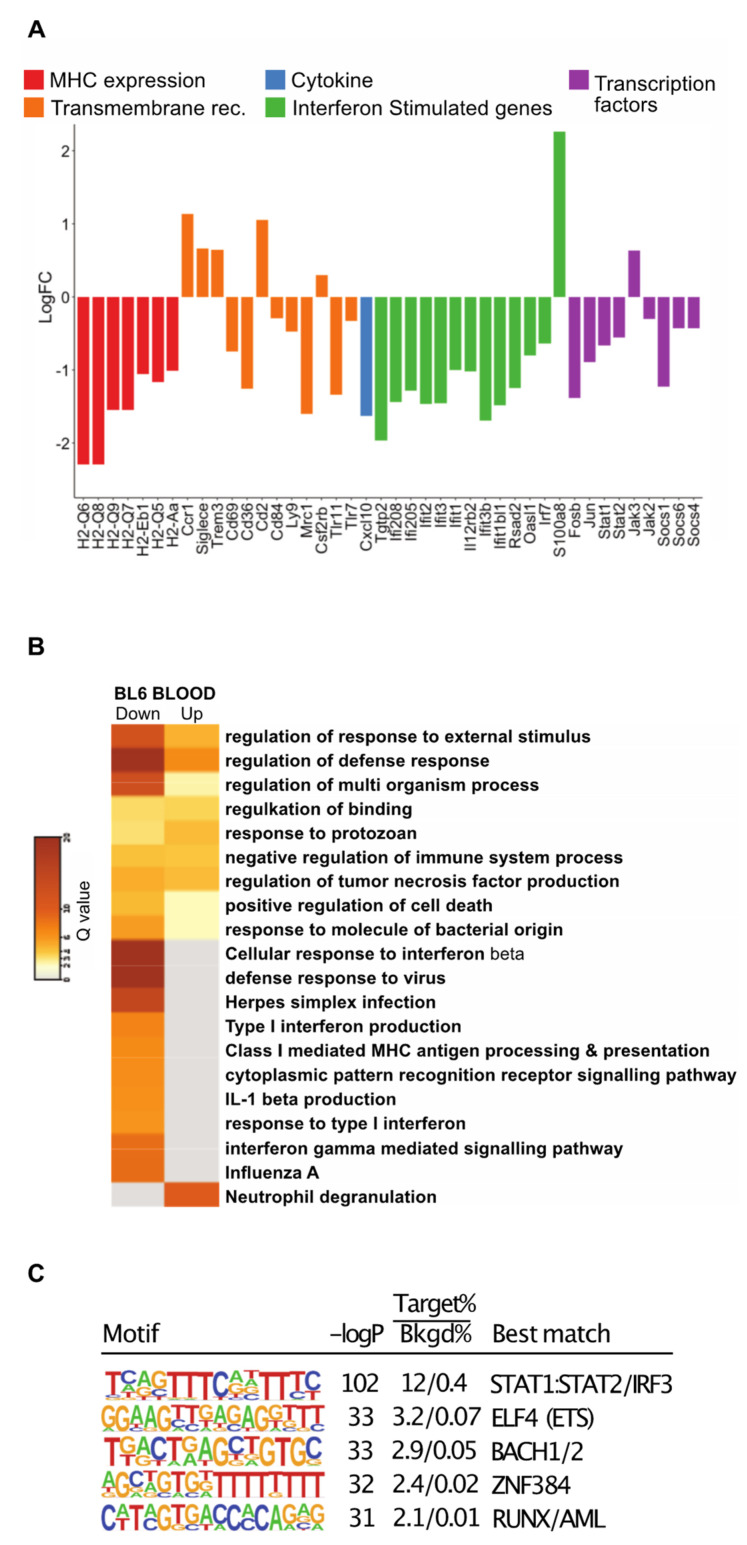
Analysis of DEGs for blood monocytes in C57BL/6 mice with late cancer. (**A**) Histogram of LogFC in selected DEGs for blood monocytes. Genes have been selected and grouped functionally as cytokines (blue), transmembrane receptors related to MHC expression (red), IFN response (orange), interferon stimulated genes (green), and transcription factors (purple). (**B**) Pathways for all DEGs in the data sets using Metascape. Each column represents up- or down-regulated gene sets. Log q.values are plotted, and each bar colored according to the log q.value on a scale of 0 to 30, represented with graduating intensity from cream to dark brown. Gene Ontology or KEGG terms are annotated. (**C**) Transcription factor motif enrichment analysis of the promoters of down-regulated DEGs revels STAT1/STAT2 as the most highly enriched.

**Figure 6 cancers-14-00833-f006:**
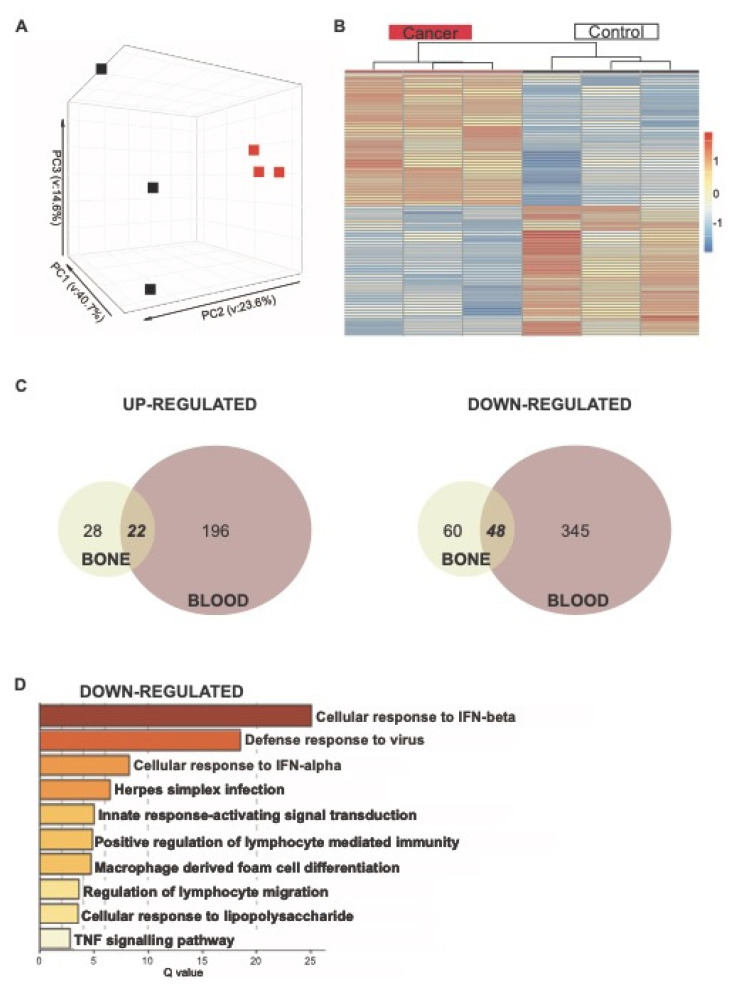
Analysis of RNA-seq of Ly6Chigh bone and blood monocytes. (**A**) PCA of Ly6c^high^ bone monocytes. (**B**) Gene expression heatmap of Ly6C^high^ bone monocytes DEGs comparing control and cancer samples with a q.value < 0.05. Samples are arranged horizontally, and sample characteristics are provided in horizontal bars for each column denoting cancer in red and control in black, with the date samples sorted in purple and blue. Genes are arranged vertically; within the heatmap, red indicates up-regulation and blue indicates down-regulation based on the tpm z-score (range (−2, 2)). Samples are clustered using complete linkage and Pearson correlation. (**C**) Venn diagram for DEGs in BM (cream) and blood (red) that are up-regulated and down-regulated in cancer. (**D**) Pathways for 47 shared down-regulated genes. Log q.values are plotted, and each bar colored according to the log q.value on a scale of 0 to 30, represented with graduating intensity from cream to dark brown. Gene Ontology or KEGG terms are annotated.

**Figure 7 cancers-14-00833-f007:**
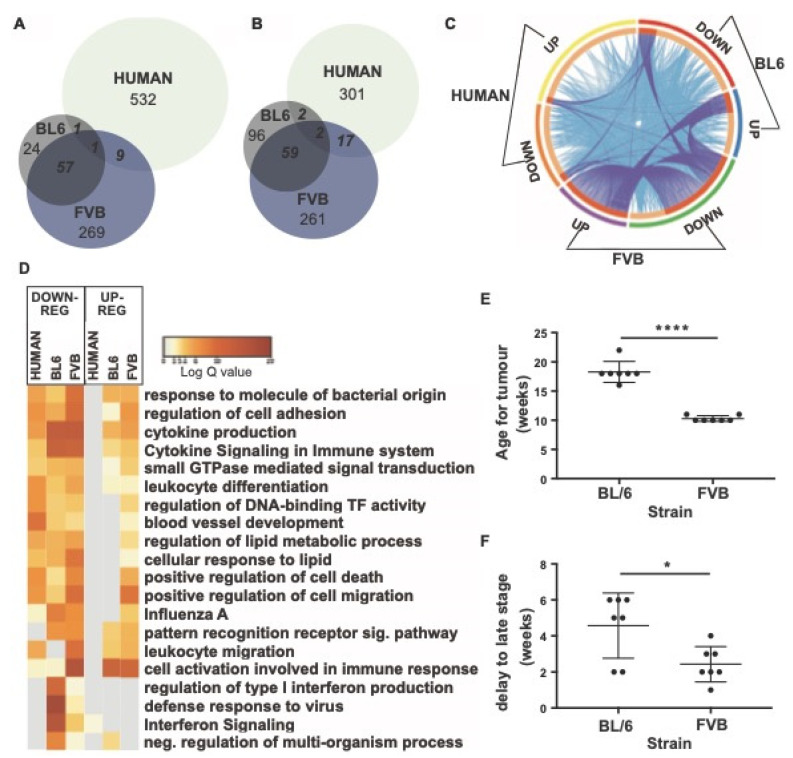
Comparison of DEGs of blood monocytes between cancer and control samples in mouse strains (FVB and C57BL/6) and humans. (**A**) Venn diagram of up-regulated DEGs in Ly6Chigh monocytes. (**B**) Venn diagram of down-regulated DEGs in Ly6Chigh monocytes. (**C**) Circos plot showing commonly enriched genes. The outer arcs represent which DEG gene list is featured: BL6: down-regulated (red arc), up-regulated (blue arc), FVB: downregulated (green arc), up-regulated (purple arc), human: down-regulated (orange arc), upregulated (yellow arc). The inner arcs reflect whether the genes are common to all three (colored in dark orange) or whether genes are common to just two groups (colored in light orange). On each occasion that a gene is in common between a group, a purple line occurs between where the gene occurs on the respective arcs. On each occasion a gene belongs to the same enriched ontology term as another gene in another group, a blue line occurs between where the gene occurs on the respective arcs. (**D**) Pathways for shared genes. Each column represents human, BL6, or FVB DEGs. Columns are grouped into pathways that were down-regulated (left) or up-regulated (right). Log q.values are plotted and each bar colored according to the log q.value on a scale of 0 to 20, represented with graduating intensity from cream to dark brown. Gene Ontology or KEGG terms are annotated. (**E**) Age of onset of tumor development in BL6 and FVB mice. (**F**) Delay in time to transition to late stage tumors. *n* = 7; * *p* value < 0.05, **** *p* value < 0.0001, paired *t*-test.

## Data Availability

Single Cell datasets are deposited in GEO: P02E01-GSE183838, P02E02-GSE184870, DEG in Appendix A.

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
