# Peer review of "Systemic Influences of Mammary Cancer on Monocytes in Mice"

_cancers, 2022, doi:10.3390/cancers14030833_

Round 1

Reviewer 1 Report

The manuscript by Robinson and colleagues nicely examines the influence of breast cancer on monocyte biology in mice. Using PyMT mouse model of breast cancer, they show there is an increased amount of CD11b+ cells in the circulating blood which, at least a good portion of it, results from increased number of monocytes and neutrophils in the blood. The authors then demonstrate that the increased number of monocytes originating from increased proliferation in the bone marrow, which is also deposited in the spleen. Although the relative abundance of classical to non-classical monocyte ratio remains the same, using single-cell and bulk RNA sequencing the author identifies differentially regulated genes (DRGs) in classical monocytes. Analysis of these genes reveals down-regulation of interferon-modulated genes, including STAT1/2, along with transcription factors Fos and Jun. Additionally, the authors confirm down-regulation of STAT1/2 regulated genes, suggesting evidence of STAT1/2 down-regulation, using transcription factor motif enrichment analysis from RNA sequencing. However, this down-regulation is not observed in Fos/Jun regulated genes. The author then investigates whether these transcriptional changes in the classical monocytes originates in the bone marrow and find 158 DRGs in the classical monocytes found inside bone marrow. Finally, the authors compare the differentially regulated genes in classical monocytes from PyMT mouse to human monocytes from breast cancer patients, revealing almost no similarities in the differentially regulated genes/pathways between the two sets. This leads to the authors' conclusion that PyMT mouse model might be insufficient to study systemic changes of human cancer.

Concerns:

1) In figure 1, the authors show that neutrophil number increases (even more than monocytes) in blood (and also bone marrow). Without characterizing and comparing the DRGs in neutrophils between PyMT mouse and cancer patients, it might be a good idea to reconsider the wording before reaching such a conclusion that "PyMT mouse model might not be sufficient to study systemic changes of human cancer".

2) In figure 1, the authors show increased number of neutrophils in blood and bone marrow. However, figure 2E suggests no increased production of neutrophil in the bone marrow. Where is the increased amount of neutophil coming from, especially considering the short-life of neutrophil?

3) It will add much value to the manuscript if the authors confirm their RNA seq data using western blot, such as decrease of STAT1/2 in the PyMT mouse compared to control. This will be especially interesting because the transcription factor motif enrichment analysis does not show  enrichment of Fos/Jun similar to STAT1/2. Thus, the authors may want to confirm that the RNA seq data aligns with the final protein expression. Additionally, functional output of these transcription factors also depends on their activation and nuclear localization which are additional consideration that the authors may want to take.

4) It is unclear why the authors reached the conclusion: "Transcriptional changes occur within the bone marrow". The presented data shows only 158 DRGs in the bone marrow monocytes compared to more than 600 in circulating blood monocytes, which is a ~4-fold increased! Thus, much more transcriptional changes are going on after the bone marrow monocytes enter circulation. Could it be a non-specific response of the monocytes against the T antigen expression since much more changes are detected in the circulatory monocytes? It would clear things up if the authors could include any control, may be induction of inflammation to show whether the DRGs are T antigen mediated change.

Minor Comments:

1) The authors performed some really good experiments and presented the data well in the manuscript. However, I was very surprised to find quite a few spelling and grammatical mistakes in the manuscript, even in the abstract section. This also includes mislabeling the figure numbers in the result section.

2) It is unclear from where the authors include the cancer patient data in figure 7. The reference they cited is in fact a review article! Did the author perform experiments with the human samples themselves? If so, it was not mentioned in the method section.

Author Response

Comments and Suggestions for Authors

The manuscript by Robinson and colleagues nicely examines the influence of breast cancer on monocyte biology in mice. Using PyMT mouse model of breast cancer, they show there is an increased amount of CD11b+ cells in the circulating blood which, at least a good portion of it, results from increased number of monocytes and neutrophils in the blood. The authors then demonstrate that the increased number of monocytes originating from increased proliferation in the bone marrow, which is also deposited in the spleen. Although the relative abundance of classical to non-classical monocyte ratio remains the same, using single-cell and bulk RNA sequencing the author identifies differentially regulated genes (DRGs) in classical monocytes. Analysis of these genes reveals down-regulation of interferon-modulated genes, including STAT1/2, along with transcription factors Fos and Jun. Additionally, the authors confirm down-regulation of STAT1/2 regulated genes, suggesting evidence of STAT1/2 down-regulation, using transcription factor motif enrichment analysis from RNA sequencing. However, this down-regulation is not observed in Fos/Jun regulated genes. The author then investigates whether these transcriptional changes in the classical monocytes originates in the bone marrow and find 158 DRGs in the classical monocytes found inside bone marrow. Finally, the authors compare the differentially regulated genes in classical monocytes from PyMT mouse to human monocytes from breast cancer patients, revealing almost no similarities in the differentially regulated genes/pathways between the two sets. This leads to the authors' conclusion that PyMT mouse model might be insufficient to study systemic changes of human cancer.

Concerns:

  • In figure 1, the authors show that neutrophil number increases (even more than monocytes) in blood (and also bone marrow). Without characterizing and comparing the DRGs in neutrophils between PyMT mouse and cancer patients, it might be a good idea to reconsider the wording before reaching such a conclusion that "PyMT mouse model might not be sufficient to study systemic changes of human cancer".

We agree and will restrict this statement to only include monocyte biology.  We will also amend the statement to recognise that other mouse models could be more representative of the human situation.

  • In figure 1, the authors show increased number of neutrophils in blood and bone marrow. However, figure 2E suggests no increased production of neutrophil in the bone marrow. Where is the increased amount of neutophil coming from, especially considering the short-life of neutrophil?

We did not explore this aspect since there is a paper from the Werb lab (Casbon et al) that specifically looked into the regulation of neutrophils in the PyMT model on an FVB background. There maybe differences between strains but this detail is beyond the current scope of the paper.  We put in the neutrophil data into our paper just to show consistency in the expansion of this populations with the previously published result.

  • It will add much value to the manuscript if the authors confirm their RNA seq data using western blot, such as decrease of STAT1/2 in the PyMT mouse compared to control. This will be especially interesting because the transcription factor motif enrichment analysis does not show  enrichment of Fos/Jun similar to STAT1/2. Thus, the authors may want to confirm that the RNA seq data aligns with the final protein expression. Additionally, functional output of these transcription factors also depends on their activation and nuclear localization which are additional consideration that the authors may want to take.

We agree this is an interesting experiment. We had attempted to perform this experiment but although we could show nuclear localisation, we were unable to show differences in groups ex vivo. Since the IFN signalling downregulation begins in the bone marrow it is probably that it is mostly complete in the blood.   These in vivo signalling pathways will be an important feature of a follow up study.

  • It is unclear why the authors reached the conclusion: "Transcriptional changes occur within the bone marrow". The presented data shows only 158 DRGs in the bone marrow monocytes compared to more than 600 in circulating blood monocytes, which is a ~4-fold increased! Thus, much more transcriptional changes are going on after the bone marrow monocytes enter circulation. Could it be a non-specific response of the monocytes against the T antigen expression since much more changes are detected in the circulatory monocytes? It would clear things up if the authors could include any control, may be induction of inflammation to show whether the DRGs are T antigen mediated change.

As the PyMT oncoprotein is express early in life the mice are fully tolerized to the PyMT antigen and do not have T cell reactions to it (our unpublished data). It is certainly possible that mutational changes that are required for progression in the PyMT model might be detected by monocytes. This could either be directly or a consequence of accumulated genetic changes, secretion of cytokines, production of exosomes etc. Indeed, in a different context we have shown with De Palma’s group that cancer produced exosomes are detected by monocytes after chemotherapy stress and this affects their ability of monocytes to home to metastatic sites and thereby enhance metastasis. This could be a possible mechanism in normal tumour development, and we have added a comment in the discussion.

We have compared inflammatory signatures in human monocytes with those found in cancer and shown no correspondence.  We do not have an equivalent data set in mice for comparison.   We have commented upon these issues in the revised manuscript.

Minor Comments:

  • The authors performed some really good experiments and presented the data well in the manuscript. However, I was very surprised to find quite a few spelling and grammatical mistakes in the manuscript, even in the abstract section. This also includes mislabeling the figure numbers in the result section.

We apologise for these oversights and have corrected the manuscript. Some of this was a mix of English and American.  We have corrected the Figure referencing.

  • It is unclear from where the authors include the cancer patient data in figure 7. The reference they cited is in fact a review article! Did the author perform experiments with the human samples themselves? If so, it was not mentioned in the method section.

The Data sets in Figure 7 come from Cassetta et al 2019 which is from the Pollard lab and has full annotation of patients.  We apologise for the mis-referencing in the text. This has been corrected.

Reviewer 2 Report

The original research article by Robinson et al is a systematic description of the changes induced in a well-defined monocytic population in MMTV-PyMT mouse model upon development of spontaneous mammary tumors. Overall the study is well designed and novel in murine context. It pursued a clear aim to provide a characterized and versatile model to explore the mechanisms behind systemic changes in circulating monocytes found in human cancer patients. However the search for the overlap of the identified genes and pathways with one available human dataset containing transcriptomes of monocytes from cancer patients provided somewhat disappointing data. Nevertheless, the study provides very interesting insights into the population counts, phenotype, half-life, origin and transcriptomic changes in the circulating as well as bone marrow monocytes of tumor-bearing mice compared to healthy age-matched counterparts.

Major points:

The study exploited one mammary cancer model (PyMT) in two murine strains (for the transcriptomics part), however the main conclusions regarding monocyte numbers, subtypes and their origin, IFN signaling involvement and enrichment for STAT1/2 TF binding sites in downregulated DEGs are derived from mice of BL6 background. Could a common signature (e.g. for IFN type I signaling or other pathways) be derived from the DEGs in monocytes of two mouse strains with PyMT mammary tumors? It would increase the impact of the findings if the Ingenuity Pathway Analysis could be performed (or similar to provide networks linking identified DEGs with upstream regulators and downstream effects). It would also be interesting to  validate some of the identified pathways from the transcriptomic analyses (e.g. IFN type I signaling).

In the Figure 7A and 7B showing the comparison of monocyte transcriptomes of human and two mouse strains, it is unclear where the numbers of DEG genes come from… In human dataset there is total 898 up and down-regulated genes that does not correspond with the number mentioned in the Results section (865). Also the number of DEGs in circulating monocytes in BL6 mice (upregulated in Venn diagram 83 and downregulated 163) was not mentioned anywhere in the results. And similarly in FVB background, the numbers in the Venn diagrams do not correspond to numbers in the Supplementary table S3. This needs to be clarified.

For the comparison of DEGs in circulating monocytes and BM monocytes in tumor-bearing mice it would be useful to show the list of the identified common genes (21 up and 47 down), e.g. as a heatmap. Could a pathway analysis be performed for set of the upregulated genes as well?

Minor comments:

  1. Abstract: rephrase the sentence: “In the bone marrow cancer does not change monocytic progenitor numbers is unaffected… “

  1. Legend for Figure 3: Number of mice used for the experiments does not fit for the panels A,B,C.

  1. Graph formatting should be identical for one image (e.g. color of error bars, thickness of axis). The use of second y axis in Figure 2A,B should be explained.

  1. TF motif enrichment analysis is not described in Methods.

Author Response

The original research article by Robinson et al is a systematic description of the changes induced in a well-defined monocytic population in MMTV-PyMT mouse model upon development of spontaneous mammary tumors. Overall the study is well designed and novel in murine context. It pursued a clear aim to provide a characterized and versatile model to explore the mechanisms behind systemic changes in circulating monocytes found in human cancer patients. However the search for the overlap of the identified genes and pathways with one available human dataset containing transcriptomes of monocytes from cancer patients provided somewhat disappointing data. Nevertheless, the study provides very interesting insights into the population counts, phenotype, half-life, origin and transcriptomic changes in the circulating as well as bone marrow monocytes of tumor-bearing mice compared to healthy age-matched counterparts.

Major points:

The study exploited one mammary cancer model (PyMT) in two murine strains (for the transcriptomics part), however the main conclusions regarding monocyte numbers, subtypes and their origin, IFN signaling involvement and enrichment for STAT1/2 TF binding sites in downregulated DEGs are derived from mice of BL6 background. Could a common signature (e.g. for IFN type I signaling or other pathways) be derived from the DEGs in monocytes of two mouse strains with PyMT mammary tumors? It would increase the impact of the findings if the Ingenuity Pathway Analysis could be performed (or similar to provide networks linking identified DEGs with upstream regulators and downstream effects). It would also be interesting to validate some of the identified pathways from the transcriptomic analyses (e.g. IFN type I signaling).

We have added data on occurrence and progression of the 2 mouse strains to show that FVB is more cancer prone than BL6 (Fig 7 E and F). We have also added appropriate references from Hunter’s group that use these genetic differences to map quantitative trait loci that affect metastasis.  We have added the list of common differentially expressed genes between mouse strains (see below); Supplemental table 4.  However, the number of genes in this list is not sufficient for pathway analysis. Nevertheless, in Fig 7D, that is the pathway analysis of all differentially expressed genes it can be seen that IFN signalling is downregulated in both mouse strains but to a lesser extent in FVB.  This maybe causal or coincidental to the change in cancer prevalence.  

In the Figure 7A and 7B showing the comparison of monocyte transcriptomes of human and two mouse strains, it is unclear where the numbers of DEG genes come from… In human dataset there is total 898 up and down-regulated genes that does not correspond with the number mentioned in the Results section (865). Also the number of DEGs in circulating monocytes in BL6 mice (upregulated in Venn diagram 83 and downregulated 163) was not mentioned anywhere in the results. And similarly in FVB background, the numbers in the Venn diagrams do not correspond to numbers in the Supplementary table S3. This needs to be clarified.

In the figure we made an error for which we apologise. This has been corrected as has the Venn diagram for the human cancers, Fig 7.

For the comparison of DEGs in circulating monocytes and BM monocytes in tumor-bearing mice it would be useful to show the list of the identified common genes (21 up and 47 down), e.g. as a heatmap. Could a pathway analysis be performed for set of the upregulated genes as well?

This has been added, supplemental Table 4.

Minor comments:

  1. Abstract: rephrase the sentence: “In the bone marrow cancer does not change monocytic progenitor numbers is unaffected… “

 Re-phrased

  1. Legend for Figure 3: Number of mice used for the experiments does not fit for the panels A,B,C.

 We have corrected this number. Apologise for the mistake.

Graph formatting should be identical for one image (e.g. color of error bars, thickness of axis). The use of second y axis in Figure 2A,B should be explained.

We have redrawn the figures for consistency

  1. TF motif enrichment analysis is not described in Methods.

Reference provided

Reviewer 3 Report

The manuscript by Robinson A et al analyzed the monocyte composition, turn-over, transcriptional profiles in a PyMT murine breast cancer model. They found that the proliferation rate of monocyte was higher in the bone marrow of tumor-bearing mice, compared to the control mice. Tumor progression did not affect the composition of monocytes in the circulation i.e. the ratios of classic monocytes versus non-classic monocytes despite the monopoiesis in bone, spleen and blood. By comparing the monocyte transcriptomes of control and tumor-bearing mice, the authors found that type I interferon signaling was down regulated in the tumor-bearing mice. It is of huge interest to find reliable cellular biomarker in the circulation for the diagnosis and prognosis of cancers. Unfortunately, the findings in mice were not translated into human patients. In general, the study will add value to the current literature.

My major concerns are:

1) Although the experiments were designed and performed well, the limitation of the study is that the conclusions were solely based on one particular virus-antigen-induced tumor model. It is unclear whether the identified type I IFN signaling pathway was due to the bias generated by this model. The authors need more discussion on this matter.

2) In Figure 1, it is better to include representative flow cytometry histograms, in addition to the summarized data. The percentages of monocytes among the whole leukocyte populations should be shown to demonstrate the monopoiesis.

3) In Figure 3, the authors observed the enhance frequencies of Ly6Chigh monocytes in the spleens of tumor-bearing mice and concluded it was due to the enhanced recruitment from blood. Could this due to enhanced retention in the spleen?

Minor issues:

In the abstract line 7: “is unaffected” should be deleted

Author Response

The manuscript by Robinson A et al analyzed the monocyte composition, turn-over, transcriptional profiles in a PyMT murine breast cancer model. They found that the proliferation rate of monocyte was higher in the bone marrow of tumor-bearing mice, compared to the control mice. Tumor progression did not affect the composition of monocytes in the circulation i.e. the ratios of classic monocytes versus non-classic monocytes despite the monopoiesis in bone, spleen and blood. By comparing the monocyte transcriptomes of control and tumor-bearing mice, the authors found that type I interferon signaling was down regulated in the tumor-bearing mice. It is of huge interest to find reliable cellular biomarker in the circulation for the diagnosis and prognosis of cancers. Unfortunately, the findings in mice were not translated into human patients. In general, the study will add value to the current literature.

My major concerns are:

  • Although the experiments were designed and performed well, the limitation of the study is that the conclusions were solely based on one particular virus-antigen-induced tumor model. It is unclear whether the identified type I IFN signaling pathway was due to the bias generated by this model. The authors need more discussion on this matter.

We agree this study is limited to one commonly used model. We did attempt to do this using orthotopic cancer models but these did not induce changes in the number of monocytes found in spontaneous transgenic models and in humans. This is probably because of the quickness of these transplant models and their lack of tumour progression from benign lesions. Thus, we were restricted to one of the few progressive and metastatic models of breast cancer.  However, we recognise the limitations and have indicated this in the discussion.  We do believe that the strain differences are important to note and we have described this in more detail.  Exploration in other models as well as in different human cancers is clearly warranted but beyond the scope of this study.

  • In Figure 1, it is better to include representative flow cytometry histograms, in addition to the summarized data. The percentages of monocytes among the whole leukocyte populations should be shown to demonstrate the monopoiesis.

We have changed Figure 1 to include these data (figure 1 A, B)

  • In Figure 3, the authors observed the enhance frequencies of Ly6Chigh monocytes in the spleens of tumor-bearing mice and concluded it was due to the enhanced recruitment from blood. Could this due to enhanced retention in the spleen?

This could indeed be the case and we have included this possibility in the discussion

Minor issues:

In the abstract line 7: “is unaffected” should be deleted

Done

Round 2

Reviewer 1 Report

The authors have presented/corrected valid reasons/arguments against almost all of the major/minor concerns. 

Reviewer 2 Report

Authors sufficiently addressed the comments in the revised version of the manuscript.